# In Situ SEM Observation of Structured Si/C Anodes Reactions in an Ionic-Liquid-Based Lithium-Ion Battery

**Huifeng Shi [1], Xianqiang Liu [1], Rui Wu [1], Yijing Zheng [2], Yonghe Li [1], Xiaopeng Cheng [1], Wilhelm Pfleging [2,3,*] and Yuefei Zhang [1,*]**

[1] Institute of Microstructure and Property of Advanced Materials, Beijing University of Technology, Beijing 100124, China; shihuifeng@emails.bjut.edu.cn (H.S.); xqliu@bjut.edu.cn (X.L.); R.Wu@emails.bjut.edu.cn (R.W.); yongheli2013@gmail.com (Y.L.); xpcheng@emails.bjut.edu.cn (X.C.)

[2] Institute for Applied Materials (IAM-AWP), Karlsruhe Institute of Technology, 76313 Eggenstein-Leopoldshafen, Karlsruhe, Germany; Yijing.zheng@kit.edu

[3] Karlsruhe Nano Micro Facility (KNMF), 76344 Eggenstein-Leopoldshafen, Germany

**Abstract:** In situ scanning electron microscopy (SEM) offers a good way to investigate the structural evolution during lithiation and delithiation processes. In this paper, the dynamical morphological evolution of 3D-line-structured/unstructured Si/C composite electrodes was observed by in situ SEM. The investigation revealed the microstructural origin of large charge capacity for 3D-line-structured anodes. Based on this proposed mechanism, a coarse optimization of 3D-line-structured anodes was proposed. These results shed light on the unique advantages of using an in situ SEM technique when studying realistic bulk batteries and designing 3D electrode structures.

**Keywords:** in situ scanning electron microscopy (SEM); lithium-ion battery; 3D structure; Si/C composites

## 1. Introduction

Among the various energy storage devices, lithium-ion batteries have been found to be one of the most promising products owing to the benefits associated with their high energy density, long lifetime cycle, and no memory effect in respect to their charge–discharge cycling [1–5]. Regarding their application in automobile industries, and especially for the future of electric vehicles (EVs), the required energy density is significantly higher than for the theoretical capacity of the traditional anode material–graphite composites. In the past four decades, several researchers have found that silicon (Si)—the second most abundant element in the earth's crust—has an exceptionally large theoretical capacity of 3579 mAh/g through forming $Li_{15}Si_4$ phase at room temperature, which is almost 10 times larger than that of graphite material [6–8]. However, pure Si anodes suffer from the large volume change (~300%) of Si when alloyed/dealloyed with Li during discharging/charging. To overcome this drawback, Si/C composites are proposed as the most promising anode materials to replace graphite in large-scale industrial production for the next generation of Li-ion batteries [9].

Except for the composition, the architecture of electrodes also has a great impact on the performance of batteries. The in-plane ultrathin microscale device using thin film electrodes usually demonstrates a higher power density but sacrifices the energy density, owing to the fast 2D ion diffusion but a low mass loading; however, a thicker electrode, i.e., with a high mass loading, conversely improves the energy density but is limited regarding the power density [5,10]. Recently [11–13], 3D micro-battery designs based on micro- and nanostructured architecture have been found to potentially be able to double the energy density without sacrificing the power density by fully utilizing the

limited space. Joong Kee Lee et al. [14] demonstrated that 3D Si/C electrodes with conformal carbon coating and empty voids were effective in accommodating the large volume changes, thus improving the electrochemical performance, i.e., cycle stability and rate capability. Utilizing an ultrashort laser radiation technique, Mangang et al. [15] successfully generated 3D surface structures into the $LiFePO_4$ cathode layer, which overcame the drawback of low lithium-ion transport in electrode materials.

However, the mechanism leading to an improved lithium-ion transport in 3D batteries is still not fully understood. Direct imaging of the electrochemical lithiation/delithiation behavior of electrode materials during the charge–discharge process with a high resolution is of great importance in designing a promising high-capacity anode architecture with enhanced electrochemical cyclability. The conventional techniques such as X-ray diffraction [16], X-ray absorption spectroscopy [17], neutron diffraction [18], and nuclear magnetic resonance [19] usually acquire only spatially averaged information. Nevertheless, observations of the structure and chemical evolution at the interface between the electrode and electrolyte surfaces using electrochemical cycling needs a high spatial resolution. In situ microscopy, especially SEM, is an ideal tool to study these issues. Compared to TEM, SEM has a significantly larger image field, which makes the experimental samples close to the realistic bulk batteries. Thus, this study reported an in situ SEM observation on the reaction of Si/C composite electrodes during lithiation and delithiation processes.

## 2. Materials and Methods

### 2.1. Materials and Preparation

The anodes were prepared by spreading the slurry on the Cu collector. Subsequently, the coated electrodes were dried through the use of a heating lid at ~60 °C in ambient air for 1 hour. The slurry was composed of 20 wt.% silicon nanoparticles with an average size of 100 nm (MTI Corporation, USA), 60 wt.% graphite (Targray Technology International Inc., USA), 10 wt.% conductive agent (Timcal Super C65, MTI Corporation, USA), 10 wt.% carboxymethylcellulose (CMC) added to styrene-butadiene rubber (SBR) (MTI corporation, USA) and poly(acrylic acid) (PAA) (Aldrich, USA). For more details refer to Reference [20].

### 2.2. In Situ SEM Experiments

In situ studies were performed using an SEM (Quanta250, FEI, USA), which was combined with an electrochemical testing instrument (BTS-4000, NEWARE, China) as shown in Figure 1a. The in situ half-cells were assembled in a glove box (Ar atmosphere, $O_2$ level < 0.1 ppm, and $H_2O$ level < 0.1 ppm). Figure 1b sketches the procedure involved in the in situ SEM observation of the Li-Si/C battery. Figure 1c presents the low-magnified SEM cross-section image of the battery, where the components of the battery were assigned using different color lines. The sandwich structure was composed of the Si/C composite (anode), separator, and pure Li foil (counter electrode). In experiments, the glass fiber separator (GF/D, Whatman) was soaked in ionic liquid electrolyte (10 wt.% bis(trifluoromethane)sulfonimide lithium salt (LiTFSI; Aladdin, China) in 1-ethyl-3-methylimidazolium bis(trifluoromethylsulfonyl)imide (EMIM TFSI; Aladdin, China). The ionic liquid that was used as an electrolyte was prepared in an argon-filled glove box and stirred for 24 h. Two kinds of Si/C composites were compared as working electrodes in our experiment. As shown in Figure 1d,e, unstructured Si/C composite and 3D-line-structured Si/C composite electrodes with a structure pitch distance of 100 μm were displayed, as processed by ultrafast laser radiation. The thickness of the Si/C composite electrode and the channel widths were 45 μm and 25 μm (Figure 1f), respectively.

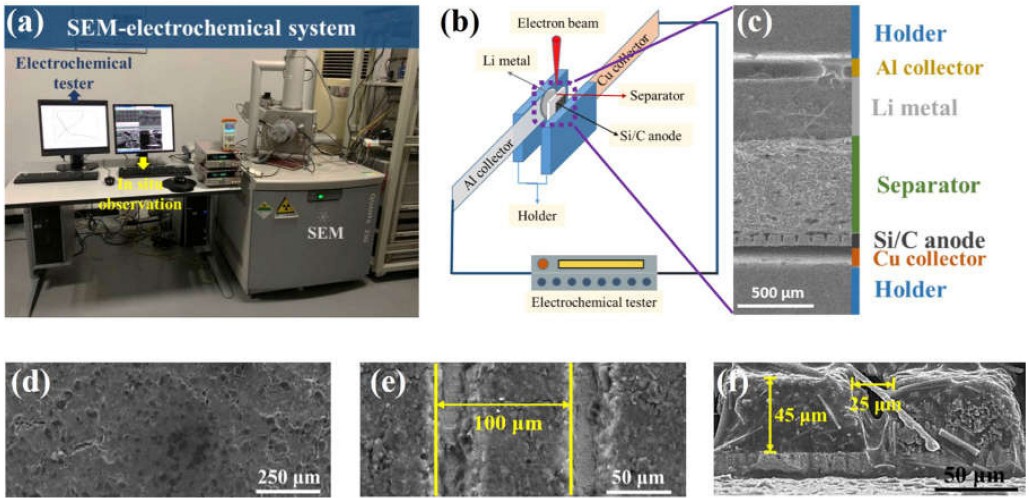

**Figure 1.** (**a**) The photograph of the SEM-electrochemical system. (**b**) Schematic drawing of in situ SEM observation for the cross-section of the Li-Si/C battery. (**c**) Low-magnification SEM image of the Li-Si/C battery for the in situ experiment. SEM images of (**d**) unstructured and (**e**) 3D-line-structured Si/C composite material. (**f**) The thickness of Si/C composites and the width of the channel.

## 3. Results

The in situ observation of morphology variation of the two types of Si/C anodes (unstructured and 3D-line-structured) was performed during the lithiation and delithiation processes (Figure 2, Videos S1 and S2). Figure 2a,c compared some of the typical SEM images of unstructured and 3D-line-structured electrodes in the in situ experiments in which the thicknesses of the electrodes were marked. Figure 2b,d present the galvanostatic charge–discharge curves of unstructured and 3D-line-structured electrodes in the in situ experiments. The images in Figure 2a,c correspond to the red points marked Figure 2b and d, respectively. The images marked with I in Figure 2a,c show the initial state of unstructured and 3D-line-structured Si/C anodes, respectively. For the unstructured electrode, the electrode began to peel off from the current collector at 56 min, as shown in image II of Figure 2a, while for the 3D-line-structured ones the channels disappeared (Figure 2c (II)) and some void-like defects appeared after 125 min (see Video S2). The images marked with III in Figure 2a,c present the morphologies of unstructured and 3D-line-structured electrodes when lithiation was finished. It is obvious that for the unstructured electrode a lift-off from the current collectors occurred, while for the laser-structured electrodes some small voids at the bottom of the composite could be detected after lithiation. In the lithiation process, the thickness of the unstructured electrode increased from 39.7 µm to 54.8 µm, and that of 3D-line-structured electrodes increased from ~46.6 (~46.1) µm to ~157.0 (~146.0) µm. Accordingly, the charge capacity of the unstructured electrode was 350 mAh/g, while that of laser-structured one was 890 mAh/g (see Figure S1). The comparison revealed that the 3D-line-structured electrodes produced a significantly improved electrochemical performance. After lithiation, a delithiation process was performed. The images marked with IV in Figure 2a,c exhibit the morphologies of electrodes after delithiation, showing that the contraction in the delithiation process was very small for both the unstructured and laser-structured electrodes, which was attributed to the short delithiation time. Figure 2b,d show the galvanostatic charge–discharge curves of unstructured and 3D-line-structured electrodes, respectively, revealing both super-low coulombic efficiency ($\approx$ 1.7%) and abnormal plateaus (0.5 V), which are due to the use of the ionic liquid as an electrolyte. The coulombic efficiency might be attributed to an irreversible EMIM cations decomposed and the plateaus can be ascribed to an irreversible EMIM cations intercalation [21,22].

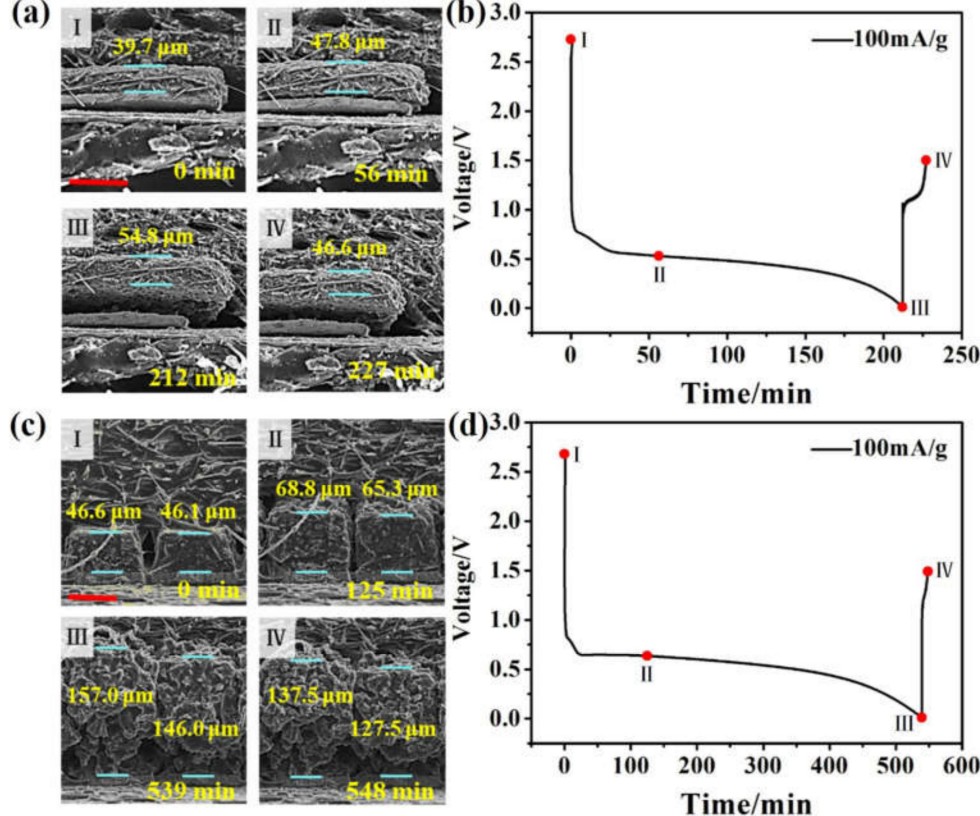

**Figure 2.** In situ SEM images of Si/C composites during charge and discharge processes. (**a**) The SEM images of unstructured Si/C composites were captured at the corresponding red points in (**b**) the galvanostatic charge–discharge curve (scale bar, 100 μm). (**c**) The SEM images of 3D-line-structured Si/C composites were captured at the corresponding red points in (**d**) the galvanostatic charge–discharge curve (scale bar, 50 μm). The curves were acquired at 100 mA/g in the voltage range of 0.01–1.5 V.

Figure 3 illustrates the lithiation process of two types of electrode materials based on in situ SEM observation. As the Figure 3a depicts, the volume of unstructured Si/C composites increased gradually during the lithiation process, resulting in most of the electrode materials peeling off from the collector at the later stage of lithiation. Figure 3b shows that the laser-generated channels were completely filled with active material due to the material expansion. As a result, the electrode materials might peel off from the collector because of the internal stress caused by the volume expansion. This is because, compared with the unstructured electrode (Figure 3a), the laser generated grooves (Figure 3b) that enlarged the contact area of electrodes and electrolyte, which allowed more lithium ions to be inserted into the electrode material and thus greatly improved its capacity. Secondly, the grooves of the 3D-line-structured anodes provided more space to release stress that volume expansion brought about. As demonstrated in our in situ experiments, the stress associated with the separation of the electrode and the current collector or defect formation resulted in the failure of these batteries.

Figure 4a shows the cross-section SEM image of 3D-line-structured Si/C anodes, in which the width and thickness of the lines can be easily measured. Because the maximum expansion in length of 3D-line-structured anodes is only about 1.35% (see Figure S2), which is very small in comparison with the maximum expansion in width and thickness (30% and 190%), we can evaluate the volume expansion in the lithiation process mainly based on the expansion in width and thickness.

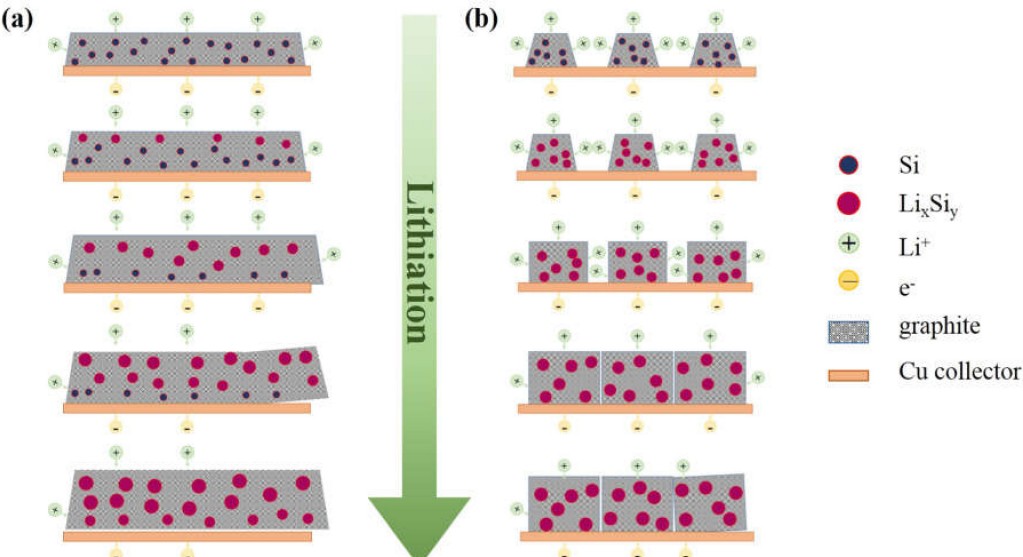

**Figure 3.** Schematic illustrations of morphology variation in (**a**) unstructured and (**b**) laser-structured Si/C composites caused by lithiation processes.

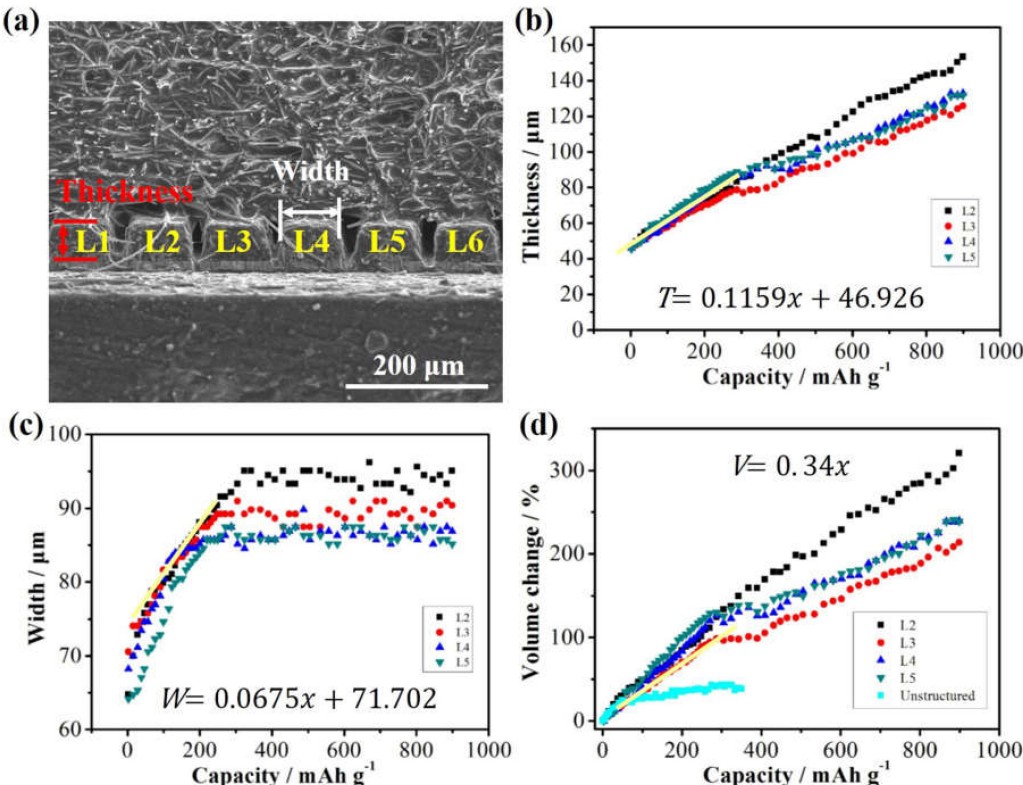

**Figure 4.** (**a**) The thickness and width of the selected sample were measured during the charging process. The (**b**) thickness and (**c**) width variations of 3D-line-structured Si/C composites during the charging process. (**d**) The volume change of 3D-line-structured Si/C composites during the charge process relating to the initial volume. (Measurement error: ± 1 μm).

Figure 4b,c show the variations in thickness and width of electrodes versus the corresponding charge capacities. In Figure 4b, the thickness of the 3D-line-structured anode exhibits a linear response to the charge capacity (lithium intercalation) before the defect formation in the contact area (see Video S2). After that, the thickness still showed a linear behavior in response to the charge capacity but with an obvious slope change due to the defect formation. The inflection point corresponded to the

charge capacity of 260 mAh/g. In Figure 4c, the width also increased linearly before colliding with 3D lines, which also corresponded to the charge capacity of 260 mAh/g. After this occurred, the width did not increase because of the space limitation. This limitation of width free expansion resulted from stress accumulation creating defects in the formation of electrode materials. Figure 4d shows the volume response of 3D-line-structured anodes as a function of the charge capacity. Combining the variations of both the thickness and width, the volume expansion also exhibited similar linear responses to the charge capacity with an inflection point at 260 mAh/g. The slope decrease after 260 mAh/g was due to the loss of conductive path caused by defects formation. Due to two-dimensional limitations on volume expansion, the unstructured anodes began to lift-off from the current collector in the early stage (56 min), thus resulting in a low charge capacity which for comparison is shown in Figure 4d.

In the first stage of the lithiation process (0–260 mAh/g), the expansion of 3D line-structured anodes has no limitation and the channels between lines released the stress in lithiation. Thus, no apparent defects were produced during this stage. This lithiation process is favored in battery applications; this indicates that the structure could be applied to batteries. The relationship between thickness (width and volume) expansion and charge capacity at this stage was expressed in the following formulae through a linear fitting analysis of Figure 4b–d:

$$T = 0.1159x + 46.926, \tag{1}$$

$$W = 0.0675x + 71.702, \tag{2}$$

$$V = 0.34x, \tag{3}$$

where x represents the charge capacity, and T, W, and V represent the thickness, width, and volume expansion, respectively. Comparing Figure 4b with Figure 4c, we found that the variation of Si/C anodes in the direction of thickness was still linear in the subsequent lithiation process, while there was no expansion in the width direction due to the space limitations. This assumes that there is enough space to release the expansion of Si, and the above linear relationships are always followed in the later stage of lithiation. The equations will be used to optimize the design of 3D-line-structured anodes.

According to the theoretical capacities of silicon (3579 mAh/g) and graphite (370 mAh/g), the theoretical capacity of Si/C composite (20 wt.% silicon nanoparticles, 60 wt.% graphite, 10 wt.% conductive agent, and 10 wt.% binder) prepared in the present study was 1172 (=0.25 × 3579 + 0.75 × 370) mAh/g. As shown in Figure 5a, when the thickness is 45 μm, the width of each line of composite material is 75 μm, and the width of the respective channel is 25 μm. However, the expansion is blocked when the lithium intercalation capacity is 260 mAh/g. A higher capacity would lead to a larger expansion of electrodes. Assuming the theoretical capacity could be achieved (meaning the electrode materials could be fully utilized as shown in Figure 5b) without structural damage, the thickness and width expansion could be evaluated using Equations (1) and (2).

$$T_{optimized} = 0.1159 \times 1172 + 46.926 = 183 \ \mu m, \tag{4}$$

$$W_{optimized} = 0.0675 \times 1172 + 71.702 = 150 \ \mu m. \tag{5}$$

As shown in Figure 5b, if the thickness of the electrode material and the width of each line structure remains unchanged, an optimized channel width might be 75 (=150–75) μm. Here, we need to point out that this evaluation was very coarse because of the penetration depth of lithium, and whether stress corresponding to a higher capacity would produce defects even without space limitations was unclear. Therefore, we define the above optimization strategy as a "suggested route to optimization". In ongoing research, further optimization will be performed.

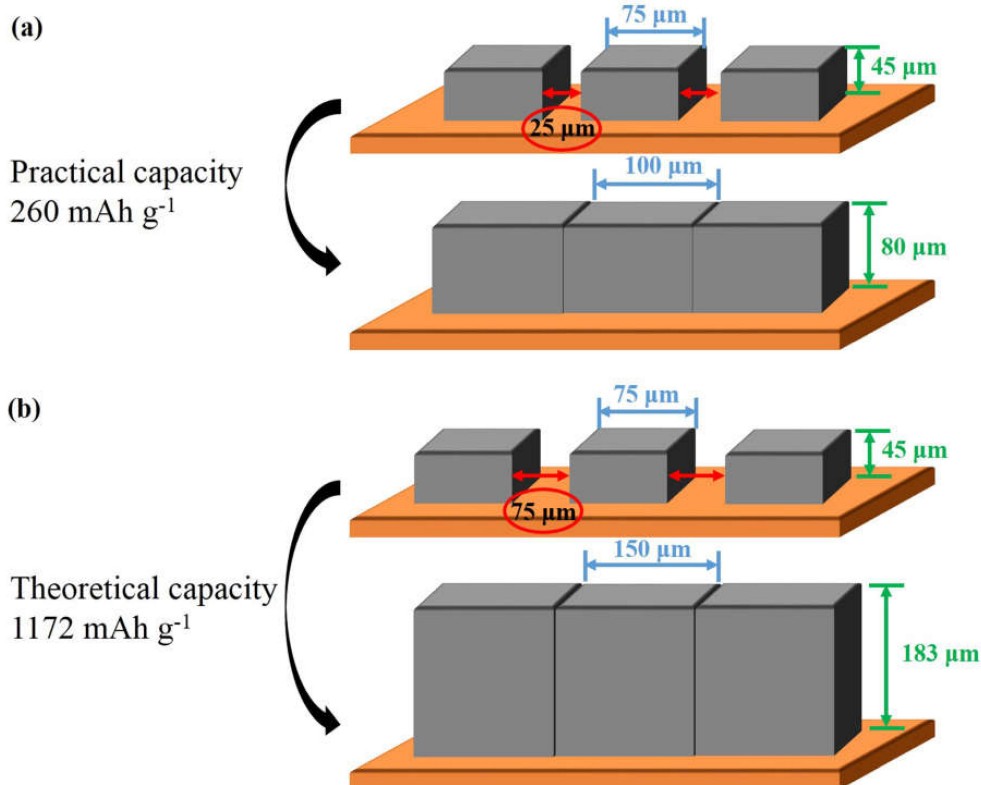

**Figure 5.** Schematic illustrations of morphology variation in (**a**) the initial structure and (**b**) the optimized structure Si/C composites before and after lithiation.

## 4. Conclusions

In summary, the lithiation process of 3D-line-structured Si/C anodes and unstructured Si/C anodes were investigated by in situ SEM techniques. The comparison revealed that through the introduction of channels to release stress brought about by volume expansion, the 3D-line-structured anodes prevent separation from the current collector. Moreover, the channels enlarged the contact area of the electrodes and lithium ionic liquid and improved the utilization of anode materials. This greatly increased the charge capacity of 3D-line-structured anodes. Based on the in situ experimental results, a coarse optimization of 3D-line-structured anodes was proposed. These results shed light on the unique advantages of the in situ SEM technique in studying realistic bulk batteries and in designing 3D electrode structures.

**Supplementary Materials:** The following are available online at http://www.mdpi.com/2076-3417/9/5/956/s1, Figure S1: Charge–discharge curves of the un-/line-structured Si/C composites at the rate of 100 mA/g, Figure S2: SEM images of (a) pristine and (b) lithiated Si/C composites, Video S1: Video of unstructured Si/C composites during charge and discharge processes, Video S2: Video of 3D-line-structured Si/C composites during charge and discharge processes.

**Author Contributions:** Conceptualization, Y.Z. and W.P.; methodology, H.S. and Y.L.; resources, Y.Z.; writing—original draft preparation, H.S.; writing—review and editing, X.L. and R.W.; supervision, X.C.; funding acquisition, Yuefei Z. and W.P.

**Funding:** This research was funded by the NSFC-DFG joint project (51761135129), National Natural Science Foundation of China (21676005), German Research Foundation (392322200), Beijing Natural Science Foundation (2172002), and Great Wall Scholarship Project (CIT and TCD20170306).

**Acknowledgments:** The support for laser materials processing was provided by the Karlsruhe Nano Micro Facility (KNMF, http://www.knmf.kit.edu/), a Helmholtz research infrastructure at KIT, and is gratefully acknowledged.

**Conflicts of Interest:** The authors declare no conflict of interest.

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
