# Peer review of "In Situ SEM Observation of Structured Si/C Anodes Reactions in an Ionic-Liquid-Based Lithium-Ion Battery"

_applsci, doi:10.3390/app9050956_

Round 1
Reviewer 1 Report
Manuscript applsci-445259 “Understanding the electrochemical reaction of structured Si/C composite anodes by in-situ scanning electron microscopy” Huifeng Shi, Xianqiang Liu, Rui Wu, Yijing Zheng, Yonghe Li, Xiaopeng Cheng, Wilhelm Pfleging, and Yuefei Zhang is one of the works in which the Dr. W. Pfleging’s method is used for structuring battery components using a femto-laser treatment, in this case for a Si/C composite anode. The advantage of this work is the direct visual observation of the electrode behavior during electrochemical lithiation. It will undoubtedly have practical application for research in the development of batteries.
However, I would say few words why this manuscript will not be published in this form. The comments and observations are as follows:
1. In my opinion, the title of article does not correspond to subject under study. I did not find the explanation of electrochemical mechanism of (de)lithiation of electrode constituents.
2. The figure 2 shows the curves of the galvanostatic measurements. For silicon electrode, the first lithiation is noticeably different from other cycles. Whether the Si/C composite electrodes were been cycling for some times before in-situ SEM study?
3. In my opinion, the forms of curves on figure 2 are strange, because graphite content is 60 % of electrode, it means what the lithiation of graphite particles represents about 24% of the electrochemical capacity of the electrode. But this is not noticeable on the galvanostatic charge-discharge curves [1]. Could the authors explain : why?
3. Could the authors explain : Why the delithiation process is so fast comparing with the lithiation? Why only one cycle was presented in work?
4. On figure 1f it is visible what the thickness of electrode is more than 45 um. I tried estimate the thickness using a scale bar (50 um) and it was ~64 um, and width of groove was ~20 um, instead of 25 um, indicated. I think the authors need to precise this because they used these values in their calculations.
5. The equation, described the dependence of volume change vs capacity and presented on figure 4(d) and page 6, is wrong.
6. I think it is possible to use the term “charge capacity” instead of “discharge capacity”.
7. One misspelling is on page 2, line 63 : “1 hours” instead of “1 hour.”
8. Page 2, line 80, the word “electrolyte” is missed.
This manuscript needs serious corrections to be published, but the videos are interesting.
Reference :
[1] Koffi P. C. Yao, John S. Okasinski, Kaushik Kalaga, Jonathan D. Almer, and Daniel P. Abraham. Operando Quantification of (De)Lithiation Behavior of Silicon–Graphite Blended Electrodes for Lithium-Ion Batteries. Adv. Energy Mater. 2019, 1803380.
Author Response
Detailed response was shown in Word file.

Reviewer 2 Report
English needs to be improved throughout the document. Many sentences do not make clear sense.
For example:
28: "the request energy density" may mean "the required energy density"?
39: "...inversely improves the energy density.." which does not have any meaning (maybe you mean "conversely, it improves the energy density.."?)
42: "without scarifying the power density.."...? (maybe "sacrificing"?)
47: "which overcomed the drawback".. should be "which overcame the drawback"
105: "became incompact"...??
117: the contraction in the delithiation process are very small" (should be is?)
193: "will prevent from the separation from current collector."... ?
The whole paragraph lines 98-120 needs to be re-worked to enhance clarity. At the moment it is quite confusing, even though the observed expansion and damage is clear.
The analysis in paragraph 175-188 is very simplistic and uses assumptions that are not based on current observations or previous work - you need to re-word it as a "suggested route to optimisation".
Author Response

(The authors gave the same response as above.)

Round 2
Reviewer 1 Report
Thank to authors for accepting of corrections.
I have an observation about Figure 4 (d). In my opinion, the equation of the linear fitting curve for dependence of volume change percentage vs charge capacity is not correct. I tried estimate the change of volume percentage for capacity, e.g. for 200 mAh g-1, using fitting curve shown on figure 4 (d) and I received value 0.68 %. But from picture it is clear visible what this value is more higher.
I assume the manuscript can be published with correction of this observation.
Author Response

(The authors gave the same response as above.)

Reviewer 2 Report
Improved - some minor english errors still exist (e.g. "Figures" when referring to more than one: Figures 2a and 2b")
Author Response

(The authors gave the same response as above.)
